

# Event-based wave statistics for the Baltic Sea

Jan-Victor Björkqvist[1,2], Hedi Kanarik[1], Laura Tuomi[1], Lauri Niskanen[3], Markus Kankainen[3]

[1]Finnish Meteorological Institute, Helsinki, 00560, Finland

[2]Norwegian Meteorological Institute, Bergen, 5007, Norway

[3]Natural Resources Institute Finland, Helsinki, 00790, Finland

*Correspondence to*: Jan-Victor Björkqvist (jan-victor.bjorkqvist@fmi.fi)

**Abstract.** Typical statistics, such as mean or percentiles, provide an excellent baseline for studying variations and changes in physical variables that have socioeconomic relevance. Nonetheless, they lack information on how often, and for how long, a

certain wave height is exceeded, which might be needed for practical applications, such as planning marine operations. Using a 29 year wave hindcast we determined the individual events where the significant wave height exceeded warning thresholds for Baltic Sea marine traffic (2.5 m, 4 m, and 7 m). During the summer months (JJA) the significant wave height exceeded 2.5 m less than twice a month. During the winter months (DJF) a significant wave height of 2.5 m was exceeded on average once a week in the larger Baltic Proper and Bothnian Sea sub-basins. Over 7 m events occurred roughly once

every other year in the larger sub-basins. Our case study for fish farm related operations compared two sites that are located 10 km and 30 km from the coast in the Bothnian Sea, where we determined wave events that could affect the feeding of the fish – meaning a significant wave height over 1 m. During the growth period of Rainbow trout (May–October) there were roughly twice as many possibly disruptive events at the location further offshore than at the location closer to the coast. Even at the less exposed location half of the wave events lasted more than 12 hours, with a few events in September and October

lasting around a week.

### 3.3.1 Introduction

Wind-generated sea surface waves impact safety at seas and the planning of offshore structures and activities. These safety and financial considerations, along with scientific interest, have motivated studies mapping the global wave climate with in situ and remote sensing measurements, and numerical models (Semedo et al., 2011; Young et al., 2011; Semedo et al., 2015;

Vanem et al., 2017). Wave statistics, trends, and extremes are therefore relatively well understood on a global and regional scale, but these general statistics might be too abstract for specific socioeconomic applications.

The Baltic Sea wave climate is relatively mild compared to the Oceans (e.g. Cieślikiewicz et al., 2008; Tuomi et al., 2011; Björkqvist et al., 2018), but the significant wave height has reached 8 m in storms (Soomere et al., 2008, Björkqvist et al.

2017, 2020). Traditional mean, maximum and percentile statistics have been used to determine the wave impact on sediment transport (Soomere and Viška, 2014) and the mean water level (Soomere et al. 2020), as well as the joint effects of waves





and water level variations (Hanson and Larson, 2010; Leijala et al., 2018; Kudryavtseva et al., 2020). Waves also affect structures in the sea, such as fish farms (Faltinsen and Shen, 2010; Karathanasi et al., 2022). Recommendations and best practices for the design of coastal and offshore structures typically rely on extreme values, such as 50, 100 or 250 year

significant wave heights (e.g. NSF 2003; Björkqvist et al. 2019).

The construction and maintenance of offshore structures require marine operations. While large cargo and passenger vessels can operate in the harshest wave conditions of the Baltic Sea, smaller vessels that are used for maintenance might be sensitive to lower sea states. Therefore, the planning of maintenance operations benefit from information on individual wave

events that can affect these smaller vessels. A typical question might be: how many times will a (possibly disruptive) wave event occur in May, and how long do such events typically last? This information cannot be deduced from traditional statistics.

This study aims to provide Baltic Sea wide, practically applicable, event-based wave statistics. The focus is on seasonal

wave events occurring multiple times per year, thus impacting day-to-day operations at sea. The study also includes a near shore case study applied to fish farming. This case study compares the wave conditions at two locations, of which one is further out to sea where the different pressures (i.e. societal, environmental) from the mainland are not so great and the eutrophication is lesser.

This paper is structured as follows: Chapter 3.3.2 introduces the model hindcast and the observational data, and Chapter 3.3.3. defines the statistics. Chapter 3.3.4 gives wave event statistics for the entire Baltic Sea, while Chapter 3.3.5 presents the case study with focus on the needs for fish farms. We end by discussing and concluding our findings.

### 3.3.2 Data and model accuracy

This study is based on the Copernicus Marine Service's Baltic Sea wave hindcast (product ref. no. 1, Table 1). The hourly

hindcast data covers the entire Baltic Sea with a 1 nautical mile (ca 1.85 km) resolution for the years 1993–2021. Of the available wave parameters we use the significant wave height, defined as $H_s=4m_0^{0.5}$, where $m_0$ is the variance of the wave field. The simulated $H_s$ had a bias and root-mean-squared-error (RMSE) of -0.04 m and 0.24 m when validated against in situ wave measurements from the Baltic Sea (Lindgren et al., 2020). Nonetheless, these results do not quantify the model accuracy in coastal areas. Also, buoys located in the north typically need to be recovered before the ice season, causing

measurement gaps in late autumn, winter, and early spring.

As an additional validation we compared the hindcast against the satellite L3 Significant Wave Height product (product ref. no. 2 and 3, Table 1). This validation, covering 2002–2021, was performed by collocating each along-track satellite





measurement to the closest model grid point, allowing for a time difference of at most 30 minutes. The -0.06 m bias and a

0.25 m RMSE over the whole domain was similar to those reported by Lindgren et al. (2020).

Finally, we compared the hindcast against coastal wave buoy measurements collected by the Finnish Meteorological Institute (FMI) in the eastern coast of the Bothnian Sea (61.325 °N 21.369 °E) from July 2017 to 5 January 2018 (product ref. no. 4, Table 1). The -0.04 m bias and 0.18 m RMSE suggests that the hindcast is accurate enough in our nearshore study region

(see Chapter 3.3.6).

*Table 1. CMEMS and non-CMEMS products used in this study, including information on data documentation.*

| Product ref. No. | Product ID & type | Data access | Documentation |
|---|---|---|---|
| 1 | BALTICSEA_REANALYSIS_WAV_003_015; Numerical models | EU Copernicus Marine Service Product (2020) | Quality Information Document (QUID): Lindgren et al., 2020 Product User Manual (PUM): Lindgren et al., 2021 |
| 2 | WAVE_GLO_PHY_SWH_L3_NRT_014_001; L3 Satellite observations | EU Copernicus Marine Service Product (2022) | Quality Information Document (QUID): Taburet et al., 2022 Product User Manual (PUM): Mertz et al., 2022 |
| 3 | WAVE_GLO_PHY_SWH_L3_MY_014_005; L3 Satellite observations | EU Copernicus Marine Service Product (2021) | Quality Information Document (QUID): Charles and Ollivier, 2021 Product User Manual (PUM): Husson and Charles, 2021 |
| 4 | FMI wave buoy measurements by the coast of the Bothnian Sea. Data originates from a Datawell Directional Waverider. | Upon request from FMI | N/A |

**3.3.3 Definition of wave events**

We define the first hour of a wave event as the time when a specific threshold is exceeded. The last hour of an event is when the $H_s$ again drops below this threshold. The duration of an event (in hours) is determined as the number of (hourly) data points between (and including) the start and the end points of the event. Nonetheless, two events are considered as one if they are under 12 hours apart and the $H_s$ doesn't drop below 90% of the threshold in between. For monthly statistics each





event is assigned to the month when the maximum is reached, even if the majority of the event took place during another
month.

We used the thresholds 1 m, 2.5 m, 4 m, and 7 m. The three highest values have been chosen by FMI to trigger national
wave warnings, since they are considered as potentially dangerous for ship traffic. The lowest threshold is relevant for lighter

operations at sea, since a survey found that fish farmers tend to avoid operating on the farms when the wave height exceeds 1
m (Kankainen, 2023).

### 3.3.4 Baltic Sea wide statistics

A 2.5 m $H_s$ was exceeded over ten times per year in all the sub-basins of the Baltic Sea (Fig. 1 a), with median durations
typically being between 10 and 16 hours (Fig. 1 d). A 4 m $H_s$ was still exceeded several times per year in most of the Baltic

Sea, but median durations over 10 h were rare. Previous studies have shown that a 7 m $H_s$ is reached only during strong
winds in the larger basins (e.g. Tuomi et al., 2011, Björkqvist et al. 2018). In our data the median duration of the 7 m events
exceeded 10 hours at some parts of the larger basins (Fig. 1 f). The 7 m threshold was not exceeded in the Gulf of Riga, Gulf
of Finland or the Bothnian Bay (Fig. 1 c).

The wave conditions in the Baltic Sea are seasonal, with the highest waves occurring in late autumn or early winter (Tuomi
et al., 2011). While large vessels operate in the Baltic Sea year around, many activities are limited to the calmer months
between late spring and early fall. We studied the seasonality in five points: four of them coincide with the location of FMI
operational wave buoys, and the fifth is taken in the southern Baltic Sea where the wave climate is the harshest (Fig. 2).

The $H_s$ in the Baltic Proper and Bothnian Sea exceeded 2.5 m (at least) about once per week between October and February,
but only once every 2 to 4 weeks in April to August. Even in the smaller basins (Gulf of Finland and Bothnian Bay) the $H_s$
exceeded 2.5 m around once every 1 to 2 weeks in the harsher period between October and January.

A $H_s$ of 4 m occurred on average less than once every other calendar month throughout the year in the smallest basins (Gulf

of Finland and Bay of Bothnia); for the larger basins this was true only between April and August. In the winter months
(DJF) $H_s$ exceeded 4 m on average 1–2 times per calendar month in the Baltic Proper main basin. The 4 m wave events can
last 12 to 24 hours even in the smaller basins (Fig. 2 a).

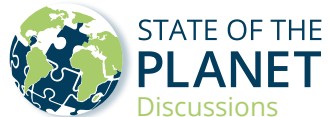

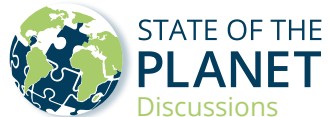

**Figure 1: Number of events per year (a–c) and median duration in hours (d–f) of cases where $H_s$ exceeds different thresholds: 2.5 m (a,d), 4 m (b,e) and 7 m (c,f). Values are calculated from 29 years hourly data from Baltic Sea wave hindcast (product ref. no. 1, Table 1).**





**Figure 2: The maximum duration of events during 1993–2021 where the significant wave height exceeded 4 m (a). The event incidence for thresholds of 2.5 m and 4 m as a function of the calendar month (b&c). The number of events exceeding 7 m during the entire 29 year period as a function of the exceedance duration in hours (d). The symbols in b)–d) refer to the areas given in a).**

A 7 m wave event was rare enough that a monthly breakdown was not meaningful. Most such events (16 events) occurred in the southern part of the Baltic Sea, with only 5 events taking place in the Northern Baltic Proper (Fig. 2 d). No 7 m wave events occurred in the Gulf of Finland and the Bothnian Bay, which is in line with existing statistics (e.g. Tuomi et al., 2011; Björkqvist et al., 2018). The 7 m wave events typically lasted under 8 hours, but one 14 h event was identified in the southeastern Baltic.





### 3.3.5 Coastal case study

Wave conditions determine what operations, if any, fish farmers can perform near the facilities. Based on a survey with fish farmers it is not feasible to feed the fish in open cage facilities when $H_s$ is over 1 m. The EU Water Framework Directive requires aquaculture facilities to meet the environmental objectives for the ecological status. This pressures fish farms to move from coastal areas, where the state is already weak, to open sea areas further offshore, where their effect on the state of the sea are reduced by the more efficient transport and mixing of nutrients. Open sea farms are exposed to harsher wave conditions, but the activities take place during the growth season for the fish (i.e. May–October for Rainbow trout) when the waves are lower.

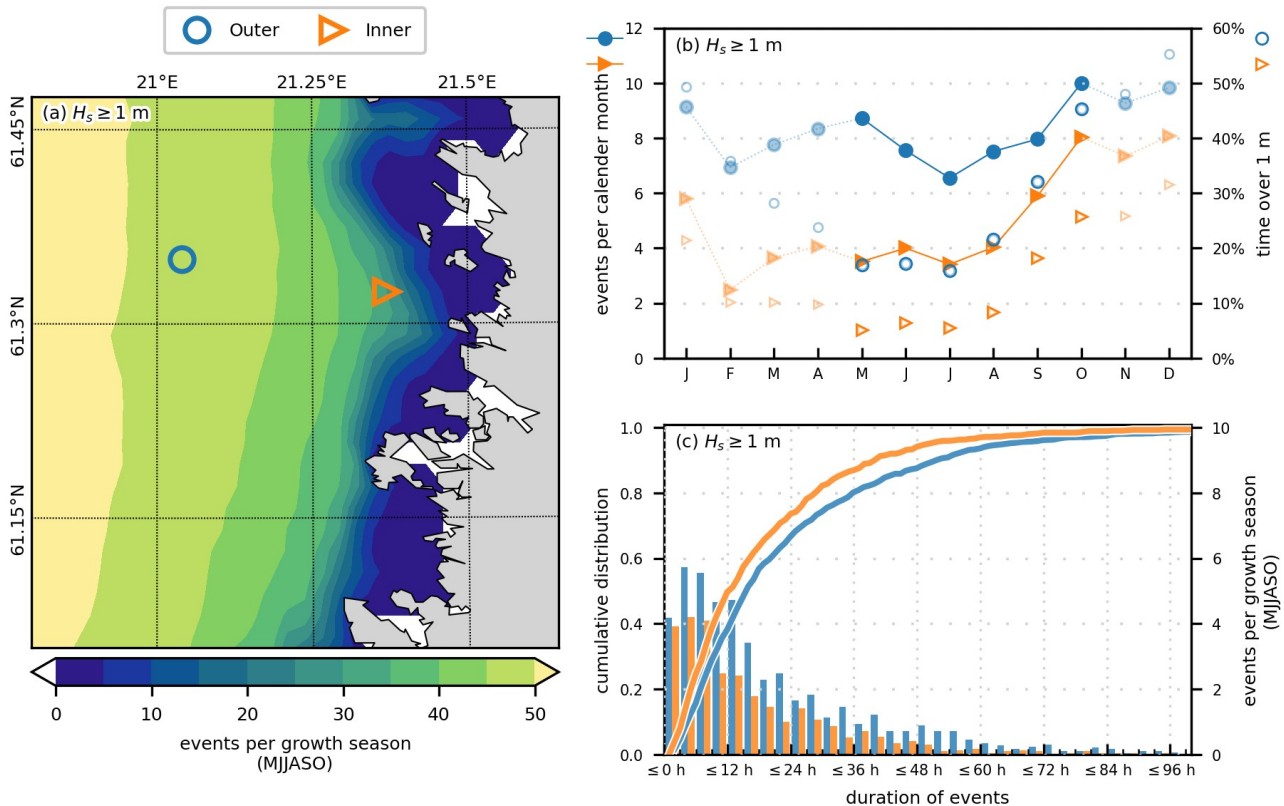

**Figure 3: Events when the significant wave height ($H_s$) exceeded 1 m during the growth season of Rainbow trout in Finland (May to October). Number of exceedances per growth season (a), event occurrence from two locations as function of calendar month (b, left axis, filled symbols) and percentage of time $H_s$ stayed over 1 m (b, right axis, empty symbols). The duration of the events during the growth season is presented as solid lines and the number of each event length is indicated as histogram (c). Bin size of histogram is 3 h so that first blue and orange bar correspond to event durations of 1, 2 and 3 h. Symbols and colours in (b,c) correspond to locations in (a).**

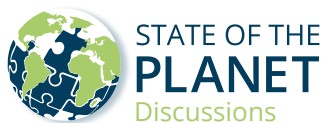

We determined events when $H_s$ exceeded 1 m for a nearshore area where fish farms are planned. We investigated the difference between two locations: one located ca 10 km from the shoreline and the second ca 20 km further out (Fig. 3). For the inner location $H_s$ exceeded 1 m around 4 times per calendar month during May–August. In September and October the
numbers were 6 and 8 times respectively – comparable to winter conditions. In the outer location $H_s$ exceeded 1 m roughly 6–9 times per calendar month during May–September. Again, the conditions in October (10 times) were comparable to November–January outside the growth season.

For the more exposed location $H_s$ exceeded 1 m a larger percent of the times during the winter than during the summer, but
this wasn't reflected in a significant increase in the number of 1 m wave events; the harsher winter wave climate simply caused longer events (Fig. 3). Half of the events lasted less than 12 h in the inner and 15 h in the outer station. Nevertheless, even during the calmer growth season 25–35% of the events lasted over 24 hours (Fig. 3c). An event lasting over 3 days is expected, on average, once every other growth season at the inner station and twice a growth season at the outer station. Still, considerably longer events are still possible: 5 events in inner station and 18 in outer station lasted over 100 h, all during
September or October. The longest event in September 1997 lasted 13 days at the outer station.

### 3.3.6 Discussion

For certain applications, such as fatigue calculations, time averaged values contain relevant information. Nonetheless, for many operations at sea – may it be heavy commercial traffic, light recreational boating, or operations at installations like fish farms – information on how often and for how long a certain wave height is exceeded might be more important than simply
knowing how many hours per year the threshold is exceeded.

A certain amount of hours per year can mean a few long events, or frequent shorter events. The type of operation, and the amount of flexibility allowed in the planning and execution, determine the impact of the event frequency and length. For example, fish farm operations with a small boat might need to be put on hold by a relatively low sea state. Nonetheless, the
flexibility in the timing will allow for adaptive execution if adequate wave forecasts exist. Larger vessels, again, can withstand higher wave heights, but typically run on a tight schedule, although with a possible slight leeway to accommodate passenger comfort, in case of passenger ferries, and fuel consumption (Jalkanen et al., 2012).

The wave conditions in the Baltic Sea depend on the season and the ice cover (e.g. Tuomi et al., 2011). Our results suggest
that wave conditions vary significantly also within one season, highlighting the value of monthly statistics. Monthly statistics are especially useful when activities span only a part of the year, with operations carried out in the beginning and end of the season.





The example used in this study is fish farming, where high waves make it difficult to feed the fish or conduct other

operations. Floating fish farms also need to be installed and removed at specific time windows in the beginning and the end

of the growth season for the fish. Moreover, all operations also depend on what kind of equipment, e.g. boat, is used. Fish

farmers install fish cages around May and remove them in September or October. The towing of the fish cages should be

done during calm sea conditions, but the process can take considerable time since it is done at a speed of only 2–4 knots.

Finding a window of possibility for towing is therefore highly relevant for the fish farmers.


The EU Water Framework Directive, and especially the ECJ Weser ruling 2015, has had an impact on where it is feasible to

grow fish. The ruling directly obligates, unless given an exemption, to reject projects that could potentially degrade the

condition of a water body or endanger the achievement of its objectives. This ruling is central in balancing between the

ecological status of the water and the different strategies to increase growth in the Blue Sector. Therefore, even if

aquaculture accounts for only 1–2 % of the total nutrient load to the Baltic Sea, this ruling has had a great impact on the

allocation of new fish farms. The directive favours broad and open sea areas where the impact of open caged fish farms on

the environment is mitigated, even though harsher environmental conditions mean less favourable conditions for practical

operations, such as towing.

Also other offshore operations might benefit from event-based statistics. For example, similar statistics can be used to plan

the construction and maintenance of offshore wind farms. The method presented here can be applied to any threshold and

variable relevant for the given operation. Furthermore, simultaneous information about, for example, the wave direction or

wave period can also be extracted. This can be used to analyse wave events from a certain direction or with a certain

steepness.

**3.3.7 Conclusions**

Based on a numerical hindcast we determined all individual wave events where the significant wave height ($H_s$) exceeded

2.5 m, 4 m, and 7 m in the Baltic Sea for the years 1993–2021. For a limited nearshore area in the Bothnian Sea we also

determined events where the significant wave height exceeded 1 m.

The number of 2.5 m and 4 m wave events were seasonal. A 7 m wave event occurred at most around 0.6 times per year on

average and only in the Baltic Proper and Bothnian Sea. The median duration of 4 m wave events was less than 10 hours in

almost the entire Baltic Sea, but even 7 m wave events that last up to 15 hours seems to be possible. Such long events are

difficult to circumvent, and while heavy marine traffic can mostly operate in any conditions in the Baltic Sea, the sea state

will affect the fuel consumption and might cause delays.

The Bothnian Sea case study targeted conditions relevant specifically for fish farming, and therefore focused on the growth season for Rainbow trout (May–October). Based on a questionnaire, a 1 m $H_s$ was found to be potentially disruptive for operations at fish farms that are typically carried out with small boats. We determined that the number of 1 m wave events in the growth season can potentially double if the fish farms are moved from 10 km to 30 km from the shore. Since open sea locations might otherwise be favoured because of e.g. nutrient loads, the challenges from harsher weather conditions needs to be balanced with regulatory requirements and environmental concerns.

The statistics found in the figures of this paper presents certain aspects of wave event statistics. Other statistics based on the given thresholds can also be easily calculated, since the single wave events have already been identified. Statistics for different thresholds require that the individual wave events for that specific threshold are first determined from the hindcast data.

**Code and data availability**

The data containing the individual wave events for the thresholds used in this paper and the Python code to determine wave events for other thresholds from the original data will be published in open repositories upon the publication of the manuscript.

**Author contributions**

The study was initiated by LT, JVB, and HK. The methodology for computing the statistics was designed by HK, JVB, and LT, and implemented by HK. HK calculated the wave statistics and performed the additional model validation. The case study was design by LN and MK in collaboration with the other authors. The original version of the manuscript was prepared by JVB and HK and revised by all authors.

**Competing interests**

The authors declare that they have no conflict of interest.

**Acknowledgements**

This work was supported by the European Union through the European Maritime and Fisheries Fund and the E.U. Copernicus Marine Service Programme.



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
