# Peer review of "Event-based wave statistics for the Baltic Sea"

_State of the Planet, 2023_

## Author Response (AR1)

Our comments and responses below are in italic

Responses to reviewer #1

General comments: The presented sort of analysis is often needed for various engineering applications; from the example of fish farms provided in the manuscript up to specification of typical length of time periods when maintenance of e.g. offshore wind generators is complicated or search and rescue operations are impossible. The results are not disruptive and largely reiterate the known features of wave climate of the Baltic Sea but are still much needed to make sense of the variability of wave fields in different parts of this water body in different seasons from the management viewpoint. This information (in particular, the duration of severe wave conditions) cannot be extracted from the classic statistical properties of the wave climate.

The results provide enough new information to deserve publication. The analysis is sound and professional. The presentation is clear and concise almost everywhere. The use of English is appropriate. Thus, I recommend this manuscript for publication with a few minor adjustments.

*Our response: Thank you for taking the time to review our manuscript. It is much appreciated. Please find our responses to your comments below.*

Technical corrections:

Line 29: it is probably meant that measured significant wave heights have reached 8 m as numerical reconstructions signal maximum wave heights around 10 m.

*Our response: This is indeed what we meant. We will change the end of the sentence to " ,but a significant wave height of 8 m has been measured during storms"*

Line 56: the provided definition of significant wave height is correct but a little bit too cryptic for non-mathematicians. Better use 4\sqrt{Hm0} and do not use italics for numbers in formulas.

*Our response: The formula for significant wave height will be changed to use the normal square root notation, and the numbers in the formula have been changed to a normal type font.*

Line 68: probably "near the coast" is meant.

*Our response: Thank you for pointing this out. We will correct that part of the sentence to "near the coast in the eastern Bothnian Sea"*

Line 118–119: please comment whether this outcome (the majority of >7 m wave events occur in the southern Baltic Sea) is consistent with the previous analysis of statistics of extreme wave fields in the Baltic proper. Could it be related to the particular time period of simulations or to the particular wind forcing?

*Our response: We will add a few sentences to note that this placement is in line with Björkqvist et al. (2018), but differs from e.g. Tuomi et al. (2011), and Räämet and Soomere (2010).*

Line 129: the expression "the state is already weak" does not make sense.

*Our response: We will update this part to read: "This asserts a pressure to establish new fish farms in open sea areas, where their effect on the ecological status is mitigated by the more efficient transport and mixing of nutrients, instead of in coastal areas, where the ecological state of the water bodies is already weaker."*

Line 200: the first sentence contains too much jargon.

*Our response: The first sentence of the line reads: "The number of 2.5 m and 4 m wave events were seasonal.", which doesn't seem to contain any jargon. We assume that there might be a mistake and the comment is actually referring to another line, but we are unsure about which one.*

Lines 213–216 seem redundant.

*Our response: These lines will be removed from the manuscript.*

*Our note: NB! Based on response from the editor we will remove the citation to the unpublished work of Kankainen (2023) and will instead add a short section to the manuscript describing the survey.*

*References: Räämet, A. and Soomere, T.: The wave climate and its seasonal variability in the northeastern Baltic Sea, Est. J. Earth Sci., https://doi.org/10.3176/earth.2010.1.08, 59, 100–113, 2010.*

Responses to reviewer #2

General comments:

The manuscript provides information related to less common statistics of wave height in the Baltic Sea, including the duration and frequency of events in each calendar month. It considers three levels of significant wave height as thresholds for these events across the entire Baltic Sea and includes a case study on a fish farm with a one-meter significant wave height threshold. The manuscript is well-written, and the figures are appropriate, providing valuable insight into the field of wave climate. Therefore, I recommend accepting the manuscript for publication with minor revisions. Please find my comments below.

*Our response: We are grateful that you agreed to review our manuscript and thank you for your constructive comments. Please find our detailed responses below.*

Specific comments:

The model hindcast used in this study is validated against satellite measurements. However, the description of this validation is rather brief, and I believe some important details are missing. For instance, it would be helpful to include information about the number of satellite data points used for this validation and the average spatial distance between the satellite track and the model grid points.

*Our response: We will add this information to the manuscript. We used 1,246,075 points with a mean distance of 0.66 km.*

Furthermore, the model data is also validated against a coastal wave buoy in the Bothnian Sea. It is worth mentioning the distance of this buoy from the coast as well as the temporal resolution of the dataset.

*Our response: We will add the following information to the paragraph regarding the wave buoy: "The buoy was located under 10 km from the coast. Data was available every 30 minutes, and we used every other data point since the model data is available every hour."*

Technical corrections:

**L11: a 29 year -> a 29-year**

*Our response: This has been corrected.*

**L11: after "wave hindcast" needs ","**

*Our response: A comma has been added.*

**L12, L13: for better clarity, I think at the end of "During the summer months (JJA)" needs ",". The same for "During the winter months (DJF)"**

*Our response: Commas has been added here as suggested to improve clarity.*

**L29-30: Please use a semicolon for separating different references, also the usage of comma after "et al." is forgotten. +L31 (Soomere et al. 2020) + L35 (Bjorkqvist et al. 2019) +L91 (e.g. Tuomi et al., 2011, Bjorkqvist et al. 2018)**

*Our response: Thank you for pointing this out. We have gone through and corrected the references in the text.*

**L79: Contractions are not used in formal writing.**

*Our response: Changed "doesn't" to "does not".*

**L129: What does "state" mean in this sentence?**

*Our response: We have changed this to "ecological status".*

**L150: Contractions are not used in formal writing.**

*Our response: Changed "wasn't" to "was not".*

**L175: I think it is better to say "at the beginning" since it refers to a particular time period**

*Our response: Thank you. This has been corrected.*

**L184: Is "Blue Sector" a specific terminology?**

*Our response: We have changed this sentence to: "This ruling is central in balancing between the ecological status of the water and the desire to increase sustainable growth in the marine and maritime sectors as laid out in e.g. the EU Blue Growth Strategy."*

**There is some inconsistency in the references. For instance, in the L239 the pages are separated by en dash, and in L243 hyphen is used. Also, the DOI in L243 is extra. The names of journals are used in abbreviations and sometimes in complete form (For example, L258, and L260 are in short form). Please double-check the references.**

*Our response: We have gone through the reference section. We have corrected the hyphens to en dashes, removed two cases of "DOI", and (in accordance with the house standard) changed all journals to the short form when such a form was available.*

*Our note: NB! Based on response from the editor we will remove the citation to the unpublished work of Kankainen (2023) and will instead add a short section to the manuscript describing the survey.*

---

## Referee Report (RR1)

General Comments

The manuscript provides a more detailed statistical analysis for the wave height duration and frequency in the Baltic Sea. The authors use three thresholds of significant wave height in the analysis, based on fish farms related works.

The MS presents good information include new aspects. Additionally, the authors have address previous comment and suggestions, and in my opinion only a couple of extra point can may be corrected/reconsidered before publication.

I listed my requested corrections below, by line where relevant.

Specific comments:

**Line 47: "are not so great" - please rephrase**

**Line 80: I believe you mean three and not thee**

**Line 154: "...weaker.." there is an extra dot after weaker**

**Line 185: "A certain amount of hours", maybe "...number of hours" is better?**

**Line 215: "Also" misses a comma after it**

Figure 2: Is not particularly difficult, but I think the colours of the symbols do not help on this colour map, so maybe consider changing them (or the colormap)?

---

## Author Response (AR2)

Dear Dr. Björkqvist,

I am pleased to inform you that your manuscript has been accepted for publication in SP, subject to technical corrections. Please consider the relevant suggestions in the two referee reports.
I would like to take this opportunity to thank you again for your support and cooperation.
Please note that all Referee and Editor reports, the author's response, as well as the different manuscript versions of the peer-review completion (post-discussion review of revised submission) will be published if the paper will be accepted for final publication in SP.
Kind regards,
Joanna Staneva
* * *
Report#1
The following basically technical corrections:
Line 80: "the" or "three"?
**Our response: Thank you, we meant three. This has been corrected.**
Line 115: consider saying that wave conditions in the Baltic Sea exhibit strong seasonal variations as the wording "The wave conditions in the Baltic Sea are seasonal" is quite jargon-like.
**Our response: Thank you for your suggestion. Nonetheless, we have decided to keep this sentence as it is. Our motivation for doing so is that while the word "seasonal" in itself might not carry the information that there is variations to all readers, this is evident by the second part of the sentence. Also, saying that the variations are "strong" is in itself a bit vague when not quantified. Thirdly, we have tried to use the active voice in this text, and we feel there is not sufficient reason to depart from that choice here.**
Line 226: I still have the opinion that the sentence "The number of 2.5 m and 4 m wave events were seasonal" is not really comprehensible for some readers and is hard to grasp grammatically. Consider saying, e.g., "The number of 2.5 m and 4 m wave events has strong seasonal variation" or similar.
**Our response: Thank you for your suggestion. We have changed this sentence to read "The number of 2.5 m and 4 m wave events varied by the season." This should be easier to grasp, while still being in the active voice and avoiding quantifying the variations as "strong".**

Report#2
General Comments
The manuscript provides a more detailed statistical analysis for the wave height duration and frequency in the Baltic Sea. The authors use three thresholds of significant wave height in theanalysis, based on fish farms related works.
The MS presents good information include new aspects. Additionally, the authors have address previous comment and suggestions, and in my opinion only a couple of extra point can may becorrected/reconsidered before publication.
I listed my requested corrections below, by line where relevant.

Specific comments:
**Line 47: "are not so great" - please rephrase**

**Our response: We have changed to "are weaker"**
**Line 80: I believe you mean three and not thee**
**Our response: Thank you, we did indeed mean three. This has been corrected.**
**Line 154: "...weaker.." there is an extra dot after weaker**
**Our response: Thank you for catching this. It has been removed.**
**Line 185: "A certain amount of hours", maybe "...number of hours" is better?**
**Our response: Thank you, we have changed this in accordance with your suggestion.**
**Line 215: "Also" misses a comma after it**
**Our response: We assume that line 213 (start of paragraph) was meant here. The comma has been added.**
Figure 2: Is not particularly difficult, but I think the colours of the symbols do not help on this colourmap, so maybe consider changing them (or the colormap)?
**Our response: This figure has been updated to use different colours for the symbols.**

**Other changes:**
**We have also added the citation to the code and data that is in the FMI data repository and modified the code and data availability statement.**

**We have corrected a few minor errors in the references.**